# Effects of Acetyl-L-Carnitine on Oxidative Stress in Amyotrophic Lateral Sclerosis Patients: Evaluation on Plasma Markers and Members of the Neurovascular Unit

**DOI:** 10.3390/antiox12101887

**Published:** 2023-10-20

**Authors:** Elena Grossini, Fabiola De Marchi, Sakthipriyan Venkatesan, Angelica Mele, Daniela Ferrante, Letizia Mazzini

**Affiliations:** 1Laboratory of Physiology, Department of Translational Medicine, Università del Piemonte Orientale, 28100 Novara, Italy; elena.grossini@med.uniupo.it (E.G.); sakthipriyan.venkatesan@uniupo.it (S.V.); 2ALS Center, Neurology Unit, Department of Translational Medicine, Università del Piemonte Orientale, 28100 Novara, Italy; fabiola.demarchi@uniupo.it (F.D.M.); 20011892@studenti.uniupo.it (A.M.); 3Statistic Unit, Department of Translational Medicine, Università del Piemonte Orientale, 28100 Novara, Italy; daniela.ferrante@uniupo.it

**Keywords:** astrocytes, vascular endothelial cells, glutathione, mitochondria, nitric oxide, oxidants, disease progression

## Abstract

Oxidative stress, the alteration of mitochondrial function, and the neurovascular unit (NVU), play a role in Amyotrophic Lateral Sclerosis (ALS) pathogenesis. We aimed to demonstrate the changes in the plasma redox system and nitric oxide (NO) in 32 new ALS-diagnosed patients in treatment with Acetyl-L-Carnitine (ALCAR) compared to healthy controls. We also evaluated the effects of plasma on human umbilical cord-derived endothelial vascular cells (HUVEC) and astrocytes. The analyses were performed at the baseline (T0), after three months (T1), and after six months (T2). In ALS patients at T0/T1, the plasma markers of lipid peroxidation, thiobarbituric acid reactive substances (TBARS) and 4-hydroxy nonenal (4-HNE) were higher, whereas the antioxidants, glutathione (GSH) and the glutathione peroxidase (GPx) activity were lower than in healthy controls. At T2, plasma TBARS and 4-HNE decreased, whereas plasma GSH and the GPx activity increased in ALS patients. As regards NO, the plasma levels were firmly lower at T0–T2 than those of healthy controls. Cell viability, and mitochondrial membrane potential in HUVEC/astrocytes treated with the plasma of ALS patients at T0–T2 were reduced, while the oxidant release increased. Those results, which confirmed the fundamental role of oxidative stress, mitochondrial function, and of the NVU in ALS pathogenesis, can have a double meaning, acting as disease markers at baseline and potential markers of drug effects in clinical practice and during clinical trials.

## 1. Introduction

Amyotrophic lateral sclerosis (ALS) is a neurodegenerative disease characterized by a progressive deterioration of upper and lower motorneurons (MNs), whose underlying mechanism has not yet been clarified [1,2]. Indeed, various targets, such as neurotoxic microglia, astrocytic excitotoxicity, impaired DNA repair, protein misfolding/aggregation, impaired proteolysis, mitochondrial dysfunction, and axonal transport defects, appear to be involved in the ALS pathogenesis [3,4,5]. Regarding the widely known existence of a genetic basis in about 10–20% of ALS cases [6], environmental factors could also account for its onset. In addition to age and male sex, smoking, exposure to β-N-methylamino-L-alanine, physical activity, trauma, and agricultural chemicals could play a role as directly causative or as risk factors for developing ALS [7,8].

The uncertainties relating to the pathogenetic mechanisms may account for the lack of a truly effective treatment. To date, only riluzole [2,9,10] and edaravone have been approved by the Food and Drug Administration (FDA) as therapeutic agents for ALS patients [11,12]; however, they have shown only limited benefits in slowing the disease progression. Given the lack of effective drugs and the ALS severity, the use of complementary and alternative therapies, such as special diets, nutritional integrators, and energy healing, is quite widespread [13]. Among these, the use of acetyl-L-carnitine (ALCAR) is undoubtedly relevant. ALCAR, one of the most common metabolites of carnitine in plasma and tissues [14,15], has anti-inflammatory and antioxidant properties [16]. The actions of ALCAR as an enhancer of the ATP synthesis from beta-oxidation of long-chain fatty acids and neurotransmitters like acetylcholine can explain its protective effects on the central nervous system (CNS) [17].

In addition, many experimental findings showed that ALCAR could protect mitochondria against oxidative stress [18]. With regard to this issue, ALCAR administration was able to induce mitochondrial biogenesis in hypoxic rats [19] and to increase mitochondrial mass after spinal cord injury [20]. In neurons and ALS animal models, ALCAR was found to exert protective effects through the modulation of mitochondrial function and neurotrophic activity [21]. Anti-apoptotic effect of acetyl-l-carnitine and I-carnitine in primary cultured neurons was reported, as well [22]. In relation to the clinical effects, however, only one study agrees on ALCAR supplementation in functional improvement [23].

A lack of detailed information about the ALCAR mechanisms of action can limit the clinical application. Indeed, a better understanding of the actions of ALCAR could be useful in clarifying its role in the management of patients affected by ALS. A possible target could be represented by the neurovascular unit (NVU), which is a network of vascular cells, glial cells, and neurons, whose alteration could hamper the blood–brain barrier (BBB) or blood–spinal cord barrier (BSCB), and lead to MNs damage due to harmful factors entering the CNS [24,25,26]. The existence of an altered balance between oxidants and antioxidants in the plasma of ALS patients has recently been demonstrated; moreover, plasma from ALS patients was found to exert deleterious effects on vascular endothelial cells and astrocytes in terms of cell survival, mitochondrial function, and oxidative stress [27]. It could, therefore, be hypothesized that the administration of ALCAR could modulate the plasma redox state and the response of the NVU to the unknown circulating mediating factors of the damage mentioned above.

In this study, we have therefore focused on the analysis of the plasma redox state in ALS patients treated with ALCAR, and on the effects of plasma on cell viability, mitochondrial membrane potential, nitric oxide (NO), and mitochondrial ROS (mitoROS) release by members of the NVU unit, such as vascular endothelial cells and astrocytes.

## 2. Materials and Methods

### 2.1. Patients

The study was performed on 32 consecutive patients diagnosed with ALS, according to the El Escorial criteria [28,29] at the Tertiary ALS Center at the “Maggiore della Carità University Hospital”, Novara, Italy, in the period September 2020–September 2022. The comparison was performed with an age (65.5 (54–71)) and sex-matched (3 males and 2 females) control group (n = 5) collected from unrelated healthy patients’ caregivers. The study was conducted following the Good Clinical Practice guidelines and the Declaration of Helsinki principles. The study was approved by the Hospital Ethical Committee (CE 54/17); each participant signed written informed consent for the handling of their clinical data and use of plasma samples for experimental purposes.

We adhered to the following inclusion criteria for the patients’ recruitment: (1) aged 18–75 years; (2) defined, clinically probable, and probable laboratory-supported ALS (El Escorial Criteria); (3) within 24 months of symptoms onset; (4) patients on riluzole treatment from at least one month before starting the ALCAR treatment; that is, T0; (5) patients without relevant comorbidities (e.g., other neurological, oncological, autoimmune diseases); (6) patients were able to provide informed consent or had a legally authorized representative willing to do so.

For each participant, we collected demographic and clinical features, including age at onset, sex, phenotype (spinal vs. bulbar) and diagnostic delay. We also collected the ALS Functional Rating Scale–Revised (ALSFRS-R) score, the Forced Vital Capacity percentage (FVC%), Body Mass Index (BMI), and the mutational status (including *C9orf72*, *SOD1*, *TARDBP*, and *FUS*). Data were collected from clinical records using an anonymous data form. An alpha-numeric code was randomly assigned to each patient to keep data collection and analysis anonymous. Patients received acetyl-L-carnitine as per clinical practice 3 g/day per os. The clinical and laboratory investigations were conducted at the time of recruitment before starting the ALCAR treatment (T0) and after six months of treatment (T2). For the plasmatic dosage of the oxidants/antioxidants and NO, we also collected blood after three months (T1). In particular, at each timing point, clinical data and plasma samples were collected. For healthy controls, plasma sample collection was executed at T0, only.

### 2.2. Collection of Blood Samples

We collected blood samples at 9 am in ALS patients and healthy controls who were fasting, using specific tubes (BD Vacutainer, containing sodium heparin to prevent clotting). The centrifugation of the plasma samples was performed by means of a centrifuge (Eppendorf, mod. 5702, rotor A-4-38), at 3100 rpm, 4 °C, for 10 min. Thereafter, we aliquoted plasma into 5 tubes, which were kept at −80 °C at the Physiology Laboratory, Università del Piemonte Orientale. We used the samples to quantify the redox state markers and to stimulate the human vascular endothelial cells (HUVEC) and the astrocytes. A pseudonymized condition was followed when handling plasma samples.

### 2.3. Measurements of the Plasma Oxidants/Antioxidants and NO

#### 2.3.1. Quantification of Thiobarbituric Acid Reactive Substances (TBARS) in Plasma Samples

Plasma lipid peroxidation was quantified by means of the TBARS assay Kit (Cayman Chemical, Ann Arbor, MI, USA), that examines the production of malondialdehyde (MDA) [27,30]. In order to perform this quantification, we added sodium dodecyl sulfate solution (100 µL) and Color Reagent (2 mL) to plasma samples (100 µL), which were boiled (1 h) and put on ice (10 min) to block the reaction. After centrifugation (10 min at 1600× *g*; 4 °C), 150 μL of each sample was transferred to a 96-well plate to quantify MDA, by means of a spectrophotometer (VICTOR™ X Multilabel Plate Reader; PerkinElmer; Waltham, MA, USA), at 530–540 nm excitation/emission wavelengths. A reference standard curve with the TBARS Standard was prepared to perform an accurate analysis of TBARS in the samples. The levels of TBARS were expressed as MDA (µM). We executed each measurement in triplicate.

#### 2.3.2. Quantification of the 4-Hydroxy Nonenal (4-HNE) in Plasma Samples

Plasma lipid peroxidation was also examined through the 4-HNE ELISA Kit (FINE TEST; Wuhan Fine Biotech Co., Ltd., Wuhan, China), in which the sample or standard competes with a fixed amount of 4-HNE on the solid phase supporter for sites on the Biotinylated Detection Antibody specific to 4-HNE [31]. Briefly, 50 µL standard or sample was added to each well, together with 50 µL Biotin-labeled Antibody, and left to incubate for 45 min at 37 °C. After washing the plate, 100 µL of SABC working solution was added and left to incubate for 30 min at 37 °C. After washing again, 90 µL of TMB substrate solution was added and left in incubation for 10–20 min at 37 °C. Thereafter, 50 µL stop solution was added and the reading was performed at 450 nm through a spectrophotometer (VICTOR™ X Multilabel Plate Reader). A reference standard curve with the 4-HNE Standard was prepared to perform an accurate analysis of 4-HNE in the samples. The concentration of 4-HNE was expressed as pg/mL. We executed each measurement in triplicate.

#### 2.3.3. Quantification of Glutathione (GSH) in Plasma Samples

The plasma GSH levels were measured by means of the Glutathione Assay Kit (Cayman Chemical) [27,31], which utilizes a carefully optimized enzymatic recycling method using glutathione reductase for the quantification of GSH. In this assay, the sulfhydryl group of GSH reacts with 5,5′-dithio-bis-2-nitrobenzoic acid (DTNB; Ellman’s reagent) and produces a yellow-colored 5-thio-2-nitrobenzoic acid (TNB). The mixed disulfide, GSTNB (between GSH and TNB) that is produced is reduced by the glutathione reductase to recycle the GSH and produce more TNB. The rate of TNB production is directly proportional to this recycling reaction, which, in turn, is directly proportional to the concentration of GSH in the sample. Because of the use of glutathione reductase in the Cayman GSH assay kit, both the reduced (GSH) and the oxidized glutathione (GSSG) are measured and the assay reflects the total GSH. In order to perform the GSH quantification, we deproteinated plasma samples through the addition of a meta-phosphoric acid solution in an equal volume. Thereafter, we centrifuged the samples at 2000× *g* for 2 min, then we collected the supernatants and added 50 µL /mL of TEAM reagent to increase the pH. Then, we transferred 50 µL of each sample to a 96-well plate for the quantification of GSH by means of a spectrophotometer (VICTOR™ X Multilabel Plate Reader), at 405–414 nm excitation/emission wavelengths. In order to execute an accurate analysis of GSH plasma levels (as µM), we prepared a standard curve, as suggested by the Glutathione Assay Kit, and executed each measurement in triplicate.

#### 2.3.4. Quantification of the Glutathione Peroxidase Activity (GPx) in Plasma Samples

The plasma glutathione peroxidase activity was measured by means of the Glutathione Peroxidase Assay Kit (Cayman Chemical), which measures the GPx activity via a coupled reaction with glutathione reductase [32]. Briefly, samples (20 µL) were incubated in an assay buffer (100 µL Tris-EDTA) and in a co-substrate mixture, which contained NADPH, glutathione, and glutathione reductase (50 µL). The reaction was initiated by adding 20 µL cumene hydroperoxide. After mixing for a few seconds, the reaction was read at 340 nm using a spectrophotometer (VICTOR™ X Multilabel Plate Reader) every min for five time-points. A standard curve was built to compare the GPx activity, which was expressed as nmol/min/mg protein after normalization to the total protein amount. We executed each measurement in triplicate.

#### 2.3.5. Evaluation of Plasma NO

We analyzed the plasma NO as both nitrites and nitrites/nitrates (NOx) through the Nitrate/Nitrite Fluorometric Assay Kit (Cayman Chemicals). Briefly, the plasma samples were ultrafiltered through the Amicon^®^ Ultra filter (30kDa MWCO Merck KGaA, Darmstadt, Germany) in order to remove proteins. For the NOx measurement, 10 µL of each sample was mixed with 70 µL assay buffer in a 96-well plate, followed by addition of the enzyme cofactor and of the nitrate reductase mixture. After incubation for 2 h, 2,3–diaminonaphthalene (DAN) and NaOH were added in each well and the absorbance was read through a spectrophotometer, as specified below. In order to measure the nitrites, after the addition of 10 µL of sample and 70 µL assay buffer, the DAN Reagent was immediately added in the 96-well plate and left in incubation for 10 min. After the addition of NaOH, the plate was immediately read at the excitation wavelength of 365 nm and emission wavelength of 430 nm through a spectrophotometer (VICTOR™ X Multilabel Plate Reader). Standard curves were built to compare the nitrites and NOx, which were expressed as μM. We executed each measurement in triplicate.

### 2.4. In Vitro Experiments

#### 2.4.1. Effects of Plasma Samples on HUVEC and Astrocytes

For the in vitro experiments, the plasma of 10 patients at T0 and T2 and of 5 healthy controls at T0 was used. Non-treated cells were also included in the analyses. In particular, we used specific Transwell inserts in order to analyze cell viability (MTT assay), mitochondrial membrane potential, mitochondrial ROS (mitoROS), and NO release in HUVEC and astrocytes treated with plasma samples (Figure 1). As in previous experiments [27], plasma samples (10% of the total volume of each insert), were placed in the apical surface of the insert for 3 h, while HUVEC cells or astrocytes were plated in the basal one. After 3 h stimulation with plasma, we removed the inserts and executed various assays, as described below. Different pools of HUVEC and astrocytes were used to perform the experiments, which were executed in triplicate and repeated three times.

#### 2.4.2. Cell Cultures

HUVEC (ATCC, catalog. no. CRL-1730TM), and mice immortalized astrocytes (kindly provided by prof. Dmitry Lim) [33], were cultured in Dulbecco’s Modified Eagle’s Medium (DMEM, Sigma-Aldrich, Milan, Italy) with 10% fetal bovine serum (FBS; Euroclone, S.p.A.; Pero, Milan, Italy), 2 mM L-glutamine (Euroclone) and 1% penicillin/streptomycin (Sigma-Aldrich).

#### 2.4.3. Cell Viability

In HUVEC and astrocytes, the viability was examined using the 1% 3-[4,5-dimethylthiazol-2-yl]-2,5-diphenyl tetrazolium bromide (MTT; Life Technologies Italia, Monza, Italy) dye [27,34,35]. To perform this analysis, 50,000 HUVEC/astrocytes/well were plated in 24-Transwells plates in complete medium (DMEM supplemented with 10% FBS). After stimulation with plasma as described above, the medium was replaced with fresh culture medium (0% red phenol and 0% FBS). Then, the MTT dye was added to the well plates and left to incubate for 2 h at 37 °C. Thereafter, the medium was replaced with an MTT solubilization solution (dimethyl sulfoxide; Sigma) and mixed in order to dissolve the formazan crystals. In each sample, the absorbance was read at 570 nm through a spectrophotometer (VICTOR™ X Multilabel Plate Reader). The viability of HUVEC/astrocytes was compared with that of control cells (non-treated cells), which was set as 100%.

#### 2.4.4. Mitochondrial Membrane Potential Measurement

We used the JC-1 assay in order to examine the mitochondrial membrane potential of HUVEC/astrocytes, as was the case in previous experiments [27,34,35]. Briefly, HUVEC and astrocytes (50,000 cells/well) positioned in 24-Transwells plates in complete medium were stimulated with plasma for 3 h, as described for the MTT assay. After stimulation, the medium was removed and cells were incubated for 15 min at 37 °C with the 5,51,6,61-tetrachloro-1,11,3,31 tetraethylbenzimidazolyl carbocyanine iodide (JC-1) 1X diluted in Assay Buffer 1X (Cayman Chemical). After washing twice using the Assay Buffer 1X, the red (excitation 550 nm/emission 600 nm) and green (excitation 485 nm/emission 535 nm) fluorescence was read through a spectrophotometer (VICTOR™ X Multilabel Plate Reader; PerkinElmer). Data were normalized in relation to non-treated cells, taken as control.

#### 2.4.5. MitoROS Release

The MitoROS release was examined using the Cayman’s Mitochondrial ROS Detection Assay Kit (Cayman Chemical) [27,35,36]. In particular, 50,000 HUVEC/astrocytes/well were positioned in 24-Transwells plates in complete medium and they were stimulated with plasma, as was the case for cell viability and mitochondrial membrane potential quantification. After stimulation, we blocked the reactions by replacing the culture media with 120 µL of Cell-Based Assay Buffer. Then, we aspirated the Buffer and added 100 µL of Mitochondrial ROS Detection Reagent Staining Solution to each well. After incubation at 37 °C, with protection from light for 20 min, we removed the Staining Solution and washed each well three times with 120 µL of PBS. In each sample, the absorbance was read at excitation/emission wavelengths of 480 nm and 560 nm, respectively, through a spectrophotometer (VICTOR™ X Multilabel Plate Reader). Data were normalized in relation to non-treated cells taken as control.

#### 2.4.6. NO Release

We used the Griess assay (Promega), to quantify the NO release in HUVEC and astrocytes [27,34]. To examine the NO release through the Griess assay, HUVEC and astrocytes (50,000 cells/well) were plated in 24-Transwells plates in complete medium and stimulated with plasma, as performed for MTT, JC-1 and mitoROS assays. At the end, we added an equal volume of Griess reagent in the sample’s supernatants and the reading of each sample was performed at 570 nm, through a spectrophotometer (VICTOR™ X Multilabel Plate Reader). The NO release was quantified in relation to a standard curve, and the production of NO was expressed as nitrites (μM).

### 2.5. Statistical Analysis

The Research Electronic Data Capture software (REDCap 13.7.18, Vanderbilt University, Nashville, TN, USA) was used to collect data. For each patient, the mean of the multiple measurements was considered for the analysis. The median and interquartile range (IQR) were used to summarize quantitative variables. The Mann–Whitney test was used to test the differences between two groups and the difference between two time points was evaluated using the Wilcoxon signed-rank test. The correlation between quantitative variables was evaluated using the Spearman’s correlation coefficient. A *p*-value < 0.05 was considered statistically significant. STATA software (StataCorp. 2021. Stata Statistical Software: Release 17. College Station, TX: StataCorp LLC) and Graph PAD (GraphPad 6.0 Software, San Diego, CA, USA) were used for statistical analysis.

## 3. Results

### 3.1. Clinical Data

Thirty-two patients (19 males and 13 females), with a median age of 67.00 (IQR: 58–70), were recruited for this study. The demographic and phenotypic features of enrolled patients are shown in Table 1. We did not observe any difference in sex and age between ALS patients and healthy controls (*p*-value > 0.05). At T1 (after three months), five patients left the study (due to death or loss at follow-up) and at T2, 11 patients left the study. In total, we analyzed 32 patients at T0, 27 patients at T1, and 21 patients at T2.

At T1, the median ALSFRS-R was 37 (IQR: 30.50–41.00), with an FVC% of 72 (IQR: 55–95) and BMI of 22.25 (IQR: 19.98–25.88). At T2, the median ALSFRS-R was 33 (IQR: 31.00–38.00), with an FVC% of 69 (IQR: 51–90) and BMI of 23.45 (IQR: 19.93–26.88).

### 3.2. Plasmatic Quantifications

In ALS patients at T0 and T1, the plasma TBARS and 4-HNE were higher (Figure 2A,B), whereas the plasma GSH levels and the GPx activity were lower than those found in healthy controls (Figure 2C,D). After six months of treatment with ALCAR (T2), plasma TBARS and 4-HNE decreased (Figure 2A,B) and plasma GSH and GPx activity increased in ALS patients (Figure 2C,D). Notably, regarding the lipid peroxidation markers, the TBARS levels measured at T2 were similar to those found in healthy controls at T0, whereas the 4-HNE levels were still higher.

Regarding NO evaluated as nitrites and NOx, the plasma levels were significantly lower at T0 and T1 than those measured in healthy controls. Despite the improvement observed at T2, however, they continued to be very low even 6 months after the start of ALCAR treatment (T2; Figure 2E,F).

### 3.3. In Vitro Experiments

We used the plasma of 10 ALS patients taken at T0 and T2 to stimulate HUVEC. We examined the effects on cell viability, mitochondrial membrane potential, mitoROS, and NO release compared with those elicited by plasma of five healthy controls at T0. As shown in Figure 3, plasma of ALS patients taken at T0 reduced cell viability, mitochondrial membrane potential, and NO release by HUVEC, while it increased mitoROS release. Meanwhile, the plasma of healthy controls did not have any effects on viability and mitochondria function, whereas it was able to increase NO release (Figure 3).

After 6 months of ALCAR treatment (T2), the plasma of ALS patients was still able to reduce viability and mitochondrial membrane and to increase the release of mitoROS in HUVEC compared with what was observed for the plasma of healthy controls. However, the effects were lower than those found with the plasma of the same patients at T0 (Figure 3A–C). It should be noted that the release of NO caused by ALS patients’ plasma taken at T2 in HUVEC was even lower than that observed at T0 (Figure 3D).

Similar results to those found in HUVEC were observed in astrocytes (Figure 4). In fact, also in this case the plasma of ALS patients at T0 reduced the viability and the mitochondrial membrane potential and increased the release of mitoROS. As observed in HUVEC, those effects were reduced when plasma of ALS patients at T2 was used to treat astrocytes. In the case of NO, it should be noted that while with the plasma of ALS patients at T0, we did not observe any particular effect, with plasma taken 6 months after starting ALCAR (T2), it was possible to observe an increase in the release of NO from the astrocytes (Figure 4D).

### 3.4. Clinical Correlations

No statistically significant correlation was found between clinical and demographic features (e.g., age, sex, site of onset, ALSFRS-R, FVC%, and BMI at baseline) and plasmatic values of TBARS, 4-HNE, GSH, GPx activity and NO at baseline and over time. Similarly, comparing the sporadic patients vs. a group of patients’ carriers of the *C9Orf72* mutation, we did not observe any difference between the two groups at baseline and over the course of the disease in terms of oxidative stress markers. Interestingly, at T1, we observed higher levels of TBARS and 4-HNE in fast-progressing patients compared to slow progressors (*p*-value: 0.02). No other variables were associated with functional progressions.

## 4. Discussion

This study confirms that the balance between oxidants and antioxidants is affected in ALS and demonstrates that the treatment with ALCAR can improve the altered redox state condition in ALS patients. As has already been highlighted, the levels of MDA, expressed as TBARS, and the amount of 4-HNE in the plasma of patients affected by ALS at the time of recruitment were higher than those found in the group of healthy controls. A reduction in the plasma GSH rate and of the GPx activity accompanied this observation.

It is noted that the assays we used to quantify the plasma lipid peroxidation and the amount of the antioxidants in ALS patients and healthy controls are those widely used to conduct this type of investigation [27,30,31,37,38,39,40,41,42].

Regarding MDA, its levels have been shown to predict worse clinical outcomes in patients with cardiovascular diseases, Alzheimer’s, and multiple sclerosis, and they are taken as a marker of ferroptosis [36]. Also, the TBARS assay is considered as a consistent and reproducible method for measuring MDA in biological fluids and cell lysates [43]. Notably, the procedure we followed in our study is the same used in recent papers to evaluate lipid peroxidation in plasma [37,38,39,44,45]. Moreover, we followed the same procedure to examine the plasma MDA levels in a previous study about ALS [27].

In addition, the 4-HNE levels, which we found to be increased in plasma, although with a decreasing trend in all time-points in ALS patients, have been associated with a greater clinical decline over an 18-month follow-up period [40].

Also, in this study, we used the DTNB-based spectrophotometric method for the quantification of the GSH levels, which was found to provide similar results to those of the chromatography-based method [46] and has been widely adopted to analyze the amount of plasma GSH [27,44,45,47,48].

Overall, therefore, our data confirm the previous conclusions relating to the presence of an alteration in the balance between oxidants and antioxidants in patients affected by neurodegenerative diseases, such as ALS. Hence, an increased level of protein carbonyl in the spinal cord and motor cortex of patients with sporadic ALS was shown [49,50]. Also, the activity/expression of SOD, catalase, glutathione reductase and glutathione transferase were found to be reduced in the cerebrospinal fluid or peripheral blood mononuclear cells of patients with familial or sporadic ALS [51,52,53].

Regarding GSH, which is well known to non-enzymatically react with ROS, a reduction in the GSH/oxidized glutathione (GSSG) ratio and of GSH levels has been observed in the cerebrospinal fluid of ALS patients [54]. In particular, the dysregulation of GSH homeostasis is believed to contribute to the development and progression of ALS [46,55].

In order to better investigate the level of antioxidants, in this study, we examined the activity of the GPx, as well. The results obtained confirm those relating to GSH, since the GPx activity was much lower than that found in plasma of controls at all time-points. Our data are also in agreement with previous observations found in plasma of ALS patients and in postmortem brain homogenates [42,52,56].

That alteration in the redox state could be associated with cellular dysfunction with particular regard to mitochondria. This was confirmed by the in vitro studies we performed by stimulating HUVEC and astrocytes with plasma of ALS patients. As previously found [27], the results obtained showed a reduction in viability of both cell types, which was accompanied by a decrease in mitochondrial membrane potential and an increased mitoROS release. We used the Mitochondrial ROS Detection Assay to investigate this, since we were focused on the analysis of the mitochondrial function of HUVEC and astrocytes. Those findings evidenced not only the role of unknown circulating factors capable of causing cellular damage through mitochondria dysfunction, but also of the members of the NVU in the onset of ALS.

Our data would support the knowledge that compromised mitochondria and oxidative stress could act as contributing factors for ALS pathology. Changes in mitochondria morphology have been shown in neurons and glial cells from ALS patients and animal models [57,58]. These alterations were also observed in both SOD1 and TDP43 ALS mice, indicating that they are common denominators of different genetic forms of ALS [59,60]. Those changes could lead to a cascade of events capable of altering mitochondrial respiration and ATP production, thus causing an increase in oxidative stress [61]. The fact that in our study, the reduction in the mitochondrial membrane potential of HUVEC and astrocytes was accompanied by an increase in mitoROS release corroborates the above issues.

The treatment of ALS patients with ALCAR was able to improve the redox state of ALS patients. Hence, the levels of TBARS and 4-HNE were, in fact, reduced, whereas those of GSH and the GPx activity were already increased at 3 months after the initial ALCAR administration, compared to T0. The improvement in the oxidative stress condition was maintained even after 6 months after the onset of ALCAR treatment.

The beneficial effects found in plasma of patients were also observed in vitro, where we obtained a reduction in the harmful effects elicited by plasma of ALS patients. In fact, our results showed a decrease in mitochondrial damage and of mitoROS production in HUVEC and astrocytes. It could be underlined that the data we obtained can provide information on the protective effect of ALCAR. Currently, little knowledge is available regarding its actions on astrocytes and existing research mainly refers to the damage induced by ethanol or caused by spinal cord injury [62,63].

In addition, our findings corroborate the role of ALCAR as a protective agent in ALS through the modulation of the mitochondria function and oxidative stress [18,19,20]. Moreover, the data we obtained highlight the members of the NVU as a possible pharmacological target of ALCAR, which could be relevant knowledge with regard to ALS management. Indeed, the NVU may play a role in the pathophysiology of ALS by preventing unknown circulating factors from entering the CNS [64]. Indeed, studies in animal models and ALS patients have shown the degeneration of endothelial cells and astrocytes end-feet processes surrounding microvessels [65,66]. In addition, vascular dysfunction could represent an early pathogenic event in ALS as shown in SOD1 mutant mice and rats, in which brain–blood barrier alterations were reported before MNs degeneration [24,25,26,67].

Interestingly, at T1, throughout the disease course, we observed a difference in lipid peroxidation markers (TBARS and 4-HNE) between fast-progressing patients and slow progressors, which could indicate a stronger dysfunction in the oxidative stress in patients with a more severe disease and, likely, a minor beneficial effect of the available treatment. Our data on 4-HNE are, therefore, in agreement with the previous data, which showed a correlation between the aforementioned marker and a greater disease severity [40].

The results about plasma NO, which was evaluated based on nitrites and NOx levels, are worthy of discussion. They showed that plasma nitrites and NOx levels were lower not only in ALS patients than those of the healthy controls at T0, but even at T1, which was at 3 months from the start of ALCAR treatment. An increase in those plasma values was observed at T2, but it settled at levels much lower than those observed in healthy controls.

Our data, therefore, highlighted the presence of an endothelial dysfunction in ALS patients, which was not affected by the treatment with ALCAR.

This finding was confirmed by the in vitro experiments, since the plasma of ALS patients was able to reduce the NO release by HUVEC at both T0 and, in this case, more strongly, at T2.

Those results could add information about the role of endothelium in the ALS genesis, which has not yet been well investigated. Endothelial cell degeneration and vascular leakage was reported in SOD1 mutants prior to MNs damage and neurovascular inflammatory response, indicating that this damage plays a central role in ALS initiation. Also, reduced levels of tight junction proteins ZO-1, occludin, and claudin-5 were demonstrated before ALS onset, as was a reduction in blood flow through the cervical and lumbar spinal cord in pre-symptomatic G93A SOD1 mice [67]. Finally, circulating endothelial cells were found to be reduced in ALS patients [68].

In this way, therapeutic approaches aiming to protect the endothelium either as direct cytoprotection or through eliminating microenvironment influences should be improved [68,69].

In astrocytes treated with ALS plasma, we observed an increased NO release at T2, which is in agreement with previous data about this issue [70,71,72,73]. NO is well known to act both as a mediator of physiological and neuroprotective actions and as an effector of neural damage [70] depending on its concentration, its oxidative/reductive status, cellular specificity, and the nature of downstream target molecules [74,75,76].

Even if this aspect has not been analyzed, we could hypothesize a different activation–expression of various NO synthase (NOS) isoforms in HUVEC and astrocytes in response to plasma from ALS patients, which would be related to the condition of oxidative stress and/or inflammation. It is well known, in fact, that while the endothelial NOS isoform is involved in the physiologic and low amount of NO release, the inducible NOS isoform is responsible for the increased NO production which could be observed in conditions characterized by oxidative stress/inflammation [77]. It could therefore be hypothesized that the presence of unknown factors in the plasma of ALS patients is capable of having the opposite effect on the above NOS. The fact that the modulation of NO release we have observed in HUVEC and astrocytes was not counteracted by ALCAR could represent a crucial factor in drug-protective actions. Hence, our data would show that on the one hand, ALCAR would act as a protective factor on the redox state and on the mitochondrial function of the NVU members, while on the other hand it would not be so effective at maintaining the balance in NO release in the NVU. If the maintenance of endothelial function and the regulation of NO release are crucial to the pathophysiology of ALS and patients’ management, the lack of ability of ALCAR to interfere with the aforementioned mechanisms could represent a bias.

Regarding the effects on oxidative stress and mitochondria, it was shown that ALCAR can be metabolized in neuronal mitochondria to free carnitine and acetyl-CoA [15]. The latter can be used as a substrate for lipids and neurotransmitters synthesis. Instead, free carnitine can be turned into products, such as carnitine derivatives of acyl-CoA conjugates, in the mitochondrial matrix, which could represent a valuable tool for reducing toxicity in oxidative stress conditions through the prevention of accumulation of long-chain fatty acids and long chain acyl-CoAs [78]. Furthermore, free carnitine has been reported to play a key role in the mitochondrial functions, fatty acid metabolism, and the production of ATP [79]. Moreover, many experimental findings demonstrated that ALCAR could protect mitochondria against oxidative stress. Also, ALCAR administration induced mitochondrial biogenesis in hypoxic rats and increased mitochondrial mass after spinal cord injury [15]. Considering what was reported above, our data corroborate the previous findings about the antioxidant effects of ALCAR, which are related to the modulation of mitochondrial function.

Our study has limitations: firstly, all patients were taking ALCAR and riluzole, although administration of these medications started one month before the baseline. As riluzole is the only drug approved to treat ALS in Europe, from an ethical point of view, it is not possible to hypothesize a control arm without riluzole and, similarly, in clinical practice, most of our patients routinely take ALCAR. Moreover, in this study, we used the same methods for assessing plasma oxidative stress as those adopted in the previous study, which focused on the evaluation of the same markers in ALS patients at T0 (before ALCAR treatment), in order to come to more meaningful conclusions about the effects of ALCAR treatment. However, it could be helpful to broaden the analysis of the oxidative stress markers by adding the quantification of hydrogen peroxide or superoxide. Also, the NO levels could be examined through a flow injection analysis. In addition, it could be useful to investigate the intracellular pathways implicated in plasma effects in HUVEC and astrocytes, particularly concerning NO release, and perform cross talk experiments between members of the NVU.

## 5. Conclusions

From the above analysis, we observed a fundamental role of oxidative stress, the alteration of mitochondrial function, and of the NVU in ALS pathogenesis at baseline and over time. We also reported a significant improvement in these parameters over time during the concomitant ALCAR treatment. These results can have a double meaning, acting both as disease markers at baseline and as potential markers of drug effects in clinical practice and during clinical trials.

## Figures and Tables

**Figure 1 antioxidants-12-01887-f001:**
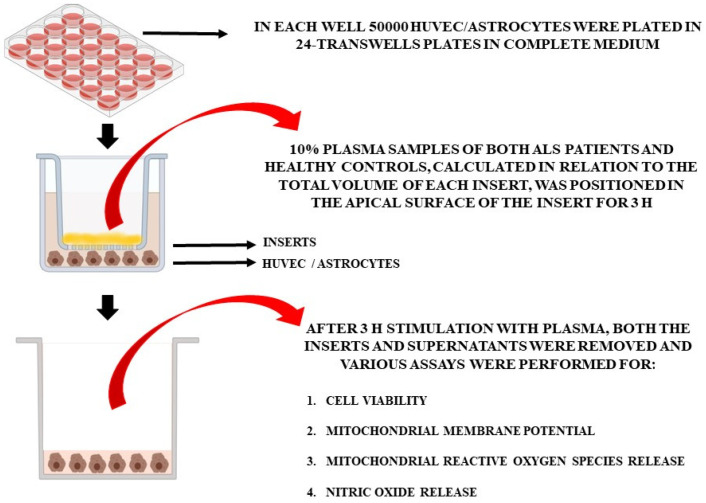
In vitro experimental protocol.

**Figure 2 antioxidants-12-01887-f002:**
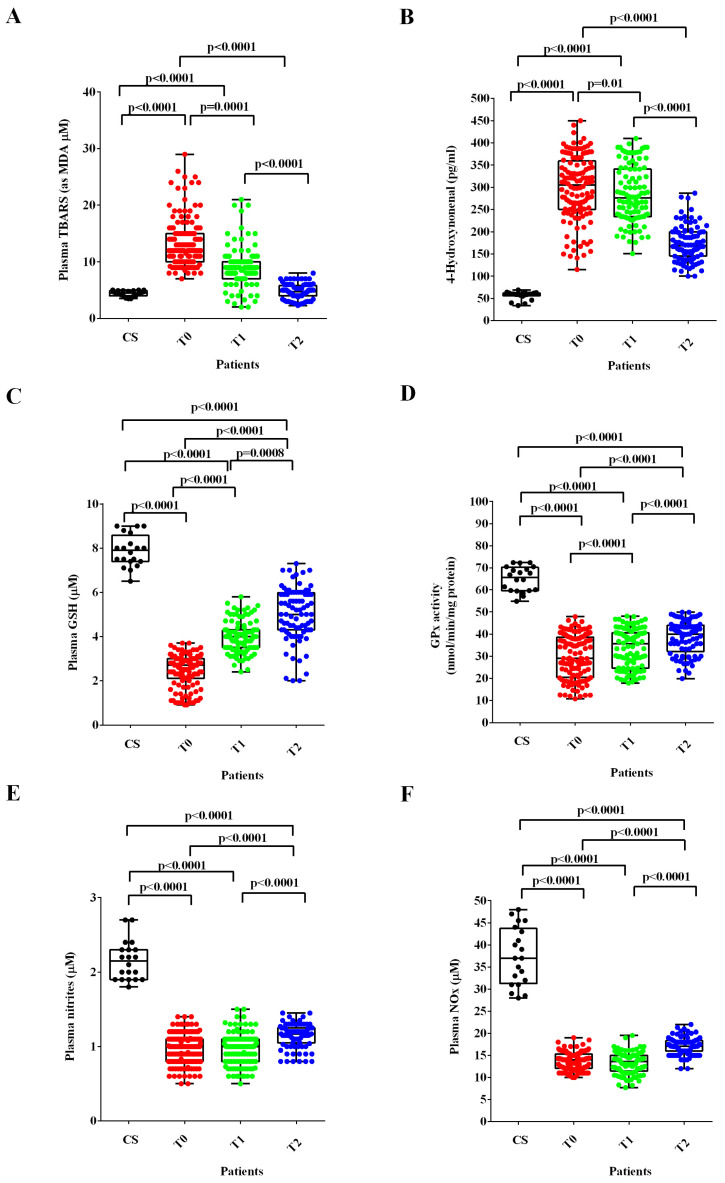
Plasma thiobarbituric acid reactive substances (TBARS), expressed as malonyldialdeide (MDA), (**A**), 4-hydroxy nonenal (4-HNE), (**B**), glutathione (GSH), (**C**), glutathione peroxidase activity (GPx), (**D**) and nitric oxide (NO) as nitrites (**E**) and nitrites/nitrates levels (NOx), (**F**) in ALS patients and healthy controls (CS) at recruitment (T0), after three months (T1) and six months (T2) Acetyl-L-Carnitine (ALCAR) treatment. Square brackets indicate significance between groups as *p*-value < 0.05.

**Figure 3 antioxidants-12-01887-f003:**
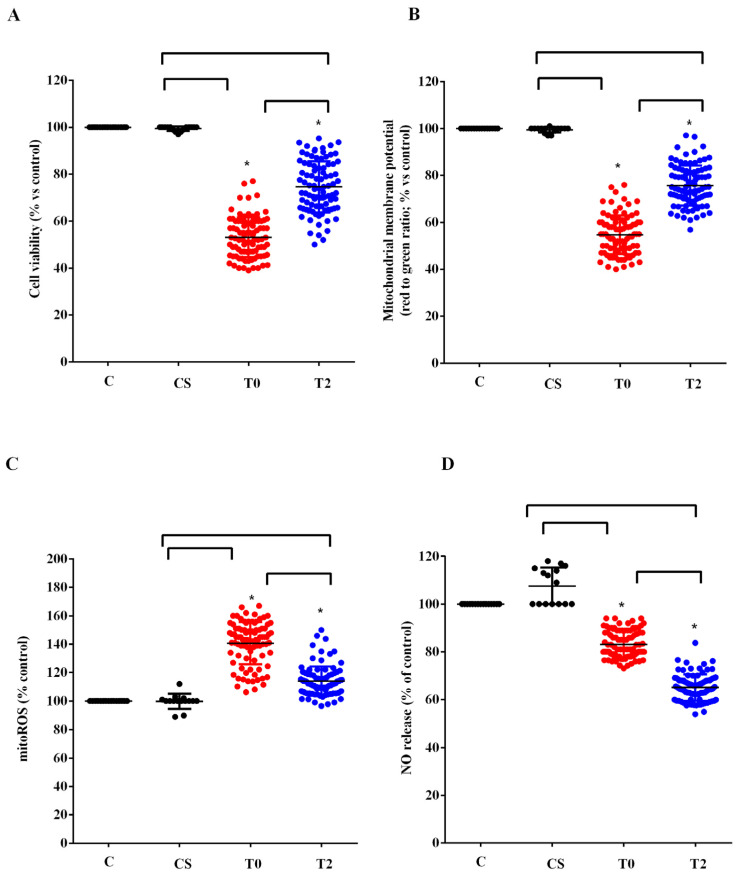
Effects of plasma of ALS patients and healthy controls (CS) on cell viability (**A**), mitochondrial membrane potential (**B**), mitochondrial reactive oxygen species (mitoROS) release (**C**), nitric oxide release (NO), (**D**) in human umbilical cord-derived endothelial vascular cells (HUVEC). T0: recruitment; T2: after six months of Acetyl-L-Carnitine (ALCAR) treatment. Square brackets indicate significance between groups as *p*-value < 0.05. * indicates *p*-value < 0.05 vs. C.

**Figure 4 antioxidants-12-01887-f004:**
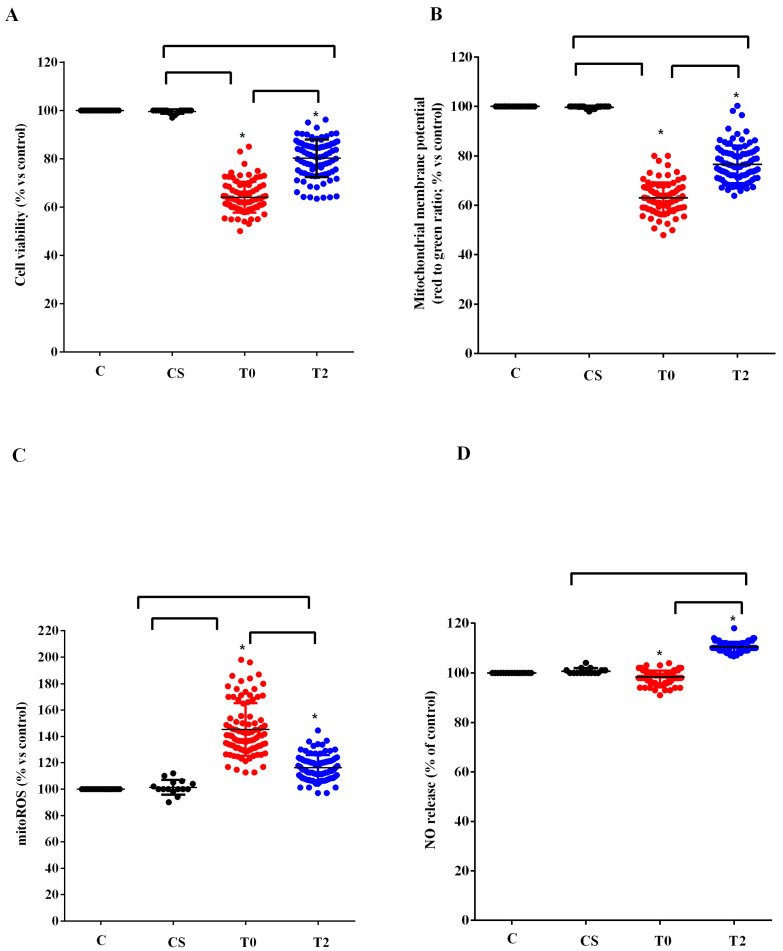
Effects of plasma of ALS patients and healthy controls (CS) on cell viability (**A**), mitochondrial membrane potential (**B**), mitochondrial reactive oxygen species (mitoROS) release (**C**), nitric oxide release (NO), (**D**) in astrocytes. T0: recruitment; T2: after six months of Acetyl-L-Carnitine (ALCAR) treatment. Square brackets indicate significance between groups as *p*-value < 0.05. * indicates *p*-value < 0.05 vs. C.

**Table 1 antioxidants-12-01887-t001:** ALS patients’ demographic and phenotypic features. ALSFRS-R: Amyotrophic Lateral Sclerosis Functional Rating Scale—Revised; FVC: forced vital capacity; BMI: body mass index. The rate of disease progression was calculated as (ALSFRS-R at baseline—last available ALSFRS-R/months) and considered fast progressors if the rate was >0.8.

Patients’ Features (n = 32)	
Sex (male/female)	19 (59%)/13 (41%)
Age at onset (median, IQR)	67 (58–70)
Phenotype (spinal/bulbar, %)	22 (69%)/10 (31%)
ALSFRS-R baseline (median, IQR)	39 (35–43)
FVC% baseline (median, IQR)	72 (49–94)
BMI baseline (median, IQR)	23.2 (20.0–26.0)
Rate of progression (fast/slow, %)	19 (59%)/13 (41%)

## Data Availability

All relevant data are available within the manuscript.

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
