# Peer review of "Effects of Acetyl-L-Carnitine on Oxidative Stress in Amyotrophic Lateral Sclerosis Patients: Evaluation on Plasma Markers and Members of the Neurovascular Unit"

_antioxidants, 2023, doi:10.3390/antiox12101887_

Round 1
Reviewer 1 Report (Previous Reviewer 1)
None
Reviewer 2 Report (Previous Reviewer 3)
The authors have included the information that I requested and so have addressed my major concerns.
While the English is understandable, it is very awkward in places so the manuscript could do with editing to make the research clearer to all readers.
This manuscript is a resubmission of an earlier submission. The following is a list of the peer review reports and author responses from that submission.
Round 1
Reviewer 1 Report
Grossini et al. Present an interesting contribution on changes in the plasma redox system and nitric oxide (NO) in new ALS-diagnosed patients in treatment with Acetyl-L-Carnitine (ALCAR) compared to healthy controls. The authors also studied the effects of plasma of ALS patients on human umbilical cord-derived endothelial vascular cells (HUVEC) and astrocytes. The authors conclude that oxidative stress-, mitochondrial function-, and neurovascular unit (NVU)-related parameters can have a role as markers of ALS pathogenesis, as well as of drugs effects.
Despite the interest of the subject, this referee has a number of concerns.
The randomized double-blind placebo-controlled trial of acetyl-L-carnitine for ALS published on 2013 showed that this molecule had options in ALS therapy (Amyotroph Lateral Scler Frontotemporal Degener. 2013 Sep;14(5-6):397-405). In fact, following these positive findings, the authors concluded that a phase III trial was needed. Based on the grim prognosis of ALS and the lack of effective treatments (still an unmet need), it is quite surprising that no clinical trial on ALCAR and ALS has been set up since then (I could not find any at www.clinicaltrial.gov). This referee would like to know why.
Regarding the methodology:
1. As clearly noted by Wade and van Rij (Clin Chem. 1989;35:336) most of problems associated with TBA assay were ignored by many researchers. Despite of these points, MDA is still used as an oxidative stress biomarker. It is now well accepted by any expert that this assay is not reliable as a marker of oxidative stress.
2. Plasma GSH levels were measured on plasma samples through the Glutathione Assay Kit 136 (Cayman Chemical). This is another methodological mistake since this assay does not accurately prevent GSH oxidation. See e.g. Anal Biochem. 1994 Mar;217(2):323-8.
3. Plasma NO were quantified in plasma samples by the Griess system and expressed as nitrite production. However, particularly in plasma samples, this methodology is not accurate. The sensitivity of the flow injection analysis, compared to the traditional Griess method, is higher by a factor of 500. It has high sensitivity, high reproducibility, and low susceptibility to interference (Nitric Oxide. 1999 Jun;3(3):225-34).
4. Why do you use the Mitochondrial ROS Detection Assay Kit, which stains mitochondria-derived ROS, and not other methods that specifically measure cellular generation of H2O2 and O2.-?
These methodological limitations must be solved to be able to reach reliable conclusions. Based on these criticisms, this contribution cannot be accepted in its present form.
I suggest to request the help of a native English speaker to review the manuscript.
Author Response
We thank the Reviewer for the critical issues raised that we used to improve the paper.
In general, the methods we use to determine the plasma redox state level in ALS patients and healthy controls are those widely used to conduct this type of investigation. We have added more sentences about this issue throughout the manuscript (discussion, page 11).
“It is to note that the assays we used to quantify the plasma lipid peroxidation and the GSH levels in ALS patients and healthy controls are those widely used to conduct this type of investigations (Angela Stufano, Camilla Isgrò, Luigi Leonardo Palese, Paolo Caretta, Luigi De Maria, Piero Lovreglio, Anna Maria Sardanelli. Oxidative Damage and Post-COVID Syndrome: A Cross-Sectional Study in a Cohort of Italian Workers Int J Mol Sci. 2023 Apr 18;24(8):7445. doi: 10.3390/ijms24087445; Afsane Bahrami, Fatemeh Nikoomanesh, Zahra Khorasanchi, Malihe Mohamadian, Gordon A Ferns. The relationship between food quality score with inflammatory biomarkers, and antioxidant capacity in young women Physiol Rep. 2023 Jan;11(2):e15590. doi: 10.14814/phy2.15590; Jousielle Márcia Dos Santos, Redha Taiar, Vanessa Gonçalves César Ribeiro, Vanessa Kelly da Silva Lage, Pedro Henrique Scheidt Figueiredo, Henrique Silveira Costa, Vanessa Pereira Lima, Borja Sañudo, Mário Bernardo-Filho, Danúbia da Cunha de Sá-Caputo, Marco Fabrício Dias Peixoto, Vanessa Amaral Mendonça, Amandine Rapin, Ana Cristina Rodrigues Lacerda. Whole-Body Vibration Training on Oxidative Stress Markers, Irisin Levels, and Body Composition in Women with Fibromyalgia: A Randomized Controlled Trial Bioengineering (Basel). 2023 Feb 16;10(2):260. doi: 10.3390/bioengineering10020260)”.
In addition, we followed the same procedures described in other papers, and we followed in a previous paper about ALS (Elena Grossini, Divya Garhwal, Sakthipriyan Venkatesan, Daniela Ferrante, Angelica Mele, Massimo Saraceno, Ada Scognamiglio, Jessica Mandrioli, Amedeo Amedei, Fabiola De Marchi, Letizia Mazzini. The Potential Role of Peripheral Oxidative Stress on the Neurovascular Unit in Amyotrophic Lateral Sclerosis Pathogenesis: A Preliminary Report from Human and In Vitro Evaluations Biomedicines. 2022 Mar 17;10(3):691. doi: 10.3390/biomedicines1003069). By this way, we could compare the results we obtained in this study with the previous ones, which concerned the levels of oxidative stress in ALS patients at T0, i.e., before starting therapy with ALCAR and come to more meaningful conclusions about the effects of ALCAR treatment on that issue (discussion, page 11).
It is to note that, for all the assays, we performed standard curves, in order to perform accurate measurements. To answer the various points in more detail:
Comment: “As clearly noted by Wade and van Rij (Clin Chem. 1989;35:336) most of problems associated with TBA assay were ignored by many researchers. Despite of these points, MDA is still used as an oxidative stress biomarker. It is now well accepted by any expert that this assay is not reliable as a marker of oxidative stress”.
Reply: despite its limited analytical specificity and ruggedness, the TBARS assay has been widely used as a generic metric of lipid peroxidation in biological fluids. It is often considered a good indicator of the levels of oxidative stress within a biological sample, provided that the sample has been properly handled and stored. The assay involves the reaction of lipid peroxidation products, primarily MDA, with thiobarbituric acid (TBA), which leads to the formation of MDA-TBA2 adducts called TBARS. TBARS yields a red-pink color that can be measured spectrophotometrically at 532 nm. The TBARS assay is performed under acidic conditions (pH = 4) and at 95 °C. Pure MDA is unstable, but these conditions allow the release of MDA from MDA bis(dimethyl acetal), which is used as the analytical standard in this method. The TBARS assay is a straightforward method that can be completed in about 2 h. The applicability of this TBARS assay is shown in human serum, low density lipoproteins, and cell lysates. The assay is consistent and reproducible, and limits of detection of 1.1 μM can be reached (Jesús Aguilar Diaz De Leon, Chad R Borges. Evaluation of Oxidative Stress in Biological Samples Using the Thiobarbituric Acid Reactive Substances Assay J Vis Exp. 2020 May 12;(159):10.3791/61122. doi: 10.3791/61122).
As added in Discussion (page 11): “As regarding the malondialdehyde, its levels have been shown to predict worse clinical outcome in patients with cardiovascular diseases, Alzheimer and multiple sclerosis and they are taken as a marker of ferroptosis (Angela Stufano, Camilla Isgrò, Luigi Leonardo Palese, Paolo Caretta, Luigi De Maria, Piero Lovreglio, Anna Maria Sardanelli. Oxidative Damage and Post-COVID Syndrome: A Cross-Sectional Study in a Cohort of Italian Workers Int J Mol Sci. 2023 Apr 18;24(8):7445. doi: 10.3390/ijms24087445) Also, the TBARS assay is considered as a consistent and reproducible method to detect malondialdehyde in biological fluids and cell lysates (Jesús Aguilar Diaz De Leon, Chad R Borges. Evaluation of Oxidative Stress in Biological Samples Using the Thiobarbituric Acid Re-active Substances Assay J Vis Exp. 2020 May 12;(159):10.3791/61122. doi: 10.3791/61122)”.
As stated above, the procedure we followed in our study is the same used in recent papers to evaluate lipid peroxidation in plasma (Angela Stufano, Camilla Isgrò, Luigi Leonardo Palese, Paolo Caretta, Luigi De Maria, Piero Lovreglio, Anna Maria Sardanelli. Oxidative Damage and Post-COVID Syndrome: A Cross-Sectional Study in a Cohort of Italian Workers Int J Mol Sci. 2023 Apr 18;24(8):7445. doi: 10.3390/ijms24087445; Afsane Bahrami, Fatemeh Nikoomanesh, Zahra Khorasanchi, Malihe Mohamadian, Gordon A Ferns. The relationship between food quality score with inflammatory biomarkers, and antioxidant capacity in young women Physiol Rep. 2023 Jan;11(2):e15590. doi: 10.14814/phy2.15590; Jousielle Márcia Dos Santos, Redha Taiar, Vanessa Gonçalves César Ribeiro, Vanessa Kelly da Silva Lage, Pedro Henrique Scheidt Figueiredo, Henrique Silveira Costa, Vanessa Pereira Lima, Borja Sañudo, Mário Bernardo-Filho, Danúbia da Cunha de Sá-Caputo, Marco Fabrício Dias Peixoto, Vanessa Amaral Mendonça, Amandine Rapin, Ana Cristina Rodrigues Lacerda. Whole-Body Vibration Training on Oxidative Stress Markers, Irisin Levels, and Body Composition in Women with Fibromyalgia: A Randomized Controlled Trial Bioengineering (Basel). 2023 Feb 16;10(2):260. doi: 10.3390/bioengineering10020260; Grossini E, Concina D, Rinaldi C, Russotto S, Garhwal D, Zeppegno P, Gramaglia C, Kul S, Panella M. Association Between Plasma Redox State/Mitochondria Function and a Flu-Like Syndrome/COVID-19 in the Elderly Admitted to a Long-Term Care Unit. Front Physiol. 2021 Dec 15;12:707587. doi: 10.3389/fphys.2021.707587. eCollection 2021; Zeppegno P, Krengli M, Ferrante D, Bagnati M, Burgio V, Farruggio S, Rolla R, Gramag-lia C, Grossini E. Psychotherapy with Music Intervention Improves Anxiety, Depression and the Redox Status in Breast Cancer Patients Undergoing Radiotherapy: A Random-ized Controlled Clinical Trial. Cancers (Basel). 2021 Apr 7;13(8):1752. doi: 10.3390/cancers13081752).
Moreover, we followed the same procedure to examine the plasma malondialdehyde levels in a previous study about ALS (Elena Grossini, Divya Garhwal, Sakthipriyan Venkatesan, Daniela Ferrante, Angelica Mele, Massimo Saraceno, Ada Scognamiglio, Jessica Mandrioli, Amedeo Amedei, Fabiola De Marchi, Letizia Mazzini. The Potential Role of Peripheral Oxidative Stress on the Neurovascular Unit in Amyotrophic Lateral Sclerosis Pathogenesis: A Preliminary Report from Human and In Vitro Evaluations Biomedicines. 2022 Mar 17;10(3):691. doi: 10.3390/biomedicines10030691).
Comment: " Plasma GSH levels were measured on plasma samples through the Glutathione Assay Kit 136 (Cayman Chemical). This is another methodological mistake since this assay does not accurately prevent GSH oxidation. See e.g. Anal Biochem. 1994 Mar;217(2):323-8”.
Reply: We have added more information about the GSH measurement (methods, page 4): “Plasma GSH levels were measured by means of the Glutathione Assay Kit (Cayman Chemical) [27,30], which utilizes a carefully optimized enzymatic recycling method, using glutathione reductase for the quantification of GSH. In this assay, the sulfhydryl group of GSH reacts with 5,5'-dithio-bis-2-nitrobenzoic acid (DTNB; Ellman’s reagent) and produces a yellow colored 5-thio-2-nitrobenzoic acid (TNB). The mixed disulfide, GSTNB (between GSH and TNB) that is produced, is reduced by the glutathione reductase to recycle the GSH and produce more TNB. The rate of TNB production is directly proportional to this recycling reaction, which, in turn, is directly proportional to the concentration of GSH in the sample. Because of the use of glutathione reductase in the Cayman GSH assay kit, both the reduced (GSH) and the oxidized glutathione (GSSG) are measured and the assay reflects the total GSH”.
Also in Discussion (page 11) we say “Also, in this study we used the DTNB-based spectrophotometric method for the quantification of the GSH levels which was found to give similar results as those of the chromatography-based method (Daniela Giustarini, Paolo Fanti, Elena Matteucci, Ranieri Rossi. Micro-method for the determination of glu-tathione in human blood. J Chromatogr B Analyt Technol Biomed Life Sci. 2014 Aug 1;964:191-4. doi: 10.1016/j.jchromb.2014.02.018. Epub 2014 Feb 19) and has been widely adopted to analyze the amount of GSH in plasma (Jayantee Kalita, Ruchi Shukla, Prakash C Pandey, Usha K Misra. Balancing between apoptosis and survival biomarkers in the patients with tuberculous meningitis Cytokine. 2022 Sep;157:155960. doi: 10.1016/j.cyto.2022.155960. Epub 2022 Jul 9; Izabella P Safe, Eduardo P Amaral, Mariana Araújo-Pereira, Marcus V G Lacerda, Vitoria S Printes, Alexandra B Souza, Francisco Beraldi-Magalhães, Wuelton M Monteiro, Vanderson S Sampaio, Beatriz Barreto-Duarte, Alice M S Andrade, Renata Spener Gomes, Allyson Guimarães Costa, Marcelo Cordeiro-Santos, Bruno B Andrade. Adjunct N-Acetylcysteine Treatment in Hospitalized Patients With HIV-Associated Tuberculosis Dampens the Oxidative Stress in Peripheral Blood: Results From the RIPENACTB Study Trial Front Immunol. 2021 Feb 4;11:602589. doi: 10.3389/fimmu.2020.602589. eCollection 2020; Elena Grossini, Divya Garhwal, Sakthipriyan Venkatesan, Daniela Ferrante, Angelica Mele, Massimo Saraceno, Ada Scognamiglio, Jessica Mandrioli, Amedeo Amedei, Fabiola De Marchi, Letizia Mazzini. The Potential Role of Peripheral Oxidative Stress on the Neurovascular Unit in Amyotrophic Lateral Sclerosis Pathogenesis: A Preliminary Report from Human and In Vitro Evaluations Biomedicines. 2022 Mar 17;10(3):691. doi: 10.3390/biomedicines10030691; Grossini E, Concina D, Rinaldi C, Russotto S, Garhwal D, Zeppegno P, Gramaglia C, Kul S, Panella M. Association Between Plasma Redox State/Mitochondria Function and a Flu-Like Syndrome/COVID-19 in the Elderly Admitted to a Long-Term Care Unit. Front Physiol. 2021 Dec 15;12:707587. doi: 10.3389/fphys.2021.707587. eCollection 2021; Zeppegno P, Krengli M, Ferrante D, Bagnati M, Burgio V, Farruggio S, Rolla R, Gramaglia C, Grossini E. Psychotherapy with Music Intervention Improves Anxiety, Depression and the Redox Status in Breast Cancer Patients Undergoing Radiotherapy: A Randomized Controlled Clinical Trial. Cancers (Basel). 2021 Apr 7;13(8):1752. doi: 10.3390/cancers13081752).
Comment: “Plasma NO were quantified in plasma samples by the Griess system and expressed as nitrite production. However, particularly in plasma samples, this methodology is not accurate. The sensitivity of the flow injection analysis, compared to the traditional Griess method, is higher by a factor of 500. It has high sensitivity, high reproducibility, and low susceptibility to interference (Nitric Oxide. 1999 Jun;3(3):225-34)”.
Reply: We added more sentences about this issue. Discussion (page 12): “ Also in this case, the plasma NO levels were determined by the Griess assay, which is widely used to perform this kind of analysis (Andrea Brizzolari, Michele Dei Cas, Danilo Cialoni, Alessandro Mar-roni, Camillo Morano, Michele Samaja, Rita Paroni, Federico Maria Rubino. High-Throughput Griess Assay of Nitrite and Nitrate in Plasma and Red Blood Cells for Human Physiology Studies under Extreme Conditions Molecules. 2021 Jul 28;26(15):4569. doi: 10.3390/molecules26154569; Maxwell B Zeigler, Emily E Fay, Sue L Moreni, Jennie Mao, Rheem A Totah, Mary F Hebert. Plasma hydrogen sulfide, nitric oxide, and thiocyanate levels are lower during pregnancy compared to postpartum in a cohort of women from the Pacific northwest of the United States Life Sci. 2023 Jun 1;322:121625. doi: 10.1016/j.lfs.2023.121625. Epub 2023 Mar 30; Ahmed Amine Zergoun, Kyle S Draleau, Faycal Chettibi, Chafia Touil-Boukoffa, Djamel Djennaoui, Taha Merghoub, Mehdi Bourouba. Plasma secretome analyses identify IL-8 and nitrites as predictors of poor prognosis in nasopharyngeal carcinoma patients Cytokine. 2022 May;153:155852. doi: 10.1016/j.cyto.2022.155852. Epub 2022 Mar 9) and we have used in our previous studies, among which about ALS, too (Elena Grossini, Claudio Molinari, Francesca Uberti, David A S G Mary, Giovanni Vacca, Philippe P Caimmi. Intracoronary melatonin increases coronary blood flow and cardiac function through β-adrenoreceptors, MT1/MT2 receptors, and nitric oxide in anesthetized pigs. J Pineal Res. 2011 Sep;51(2):246-57. doi: 10.1111/j.1600-079X.2011.00886.x. Epub 2011 May 4; Elena Grossini Giulia Raina, Serena Farruggio, Lara Camillo, Claudio Molinari, David Mary, Gillian Elisabeth Walker, Gianni Bona, Giovanni Vacca, Stefania Moia, Flavia Prodam, Daniela Surico. Intracoronary Des-Acyl Ghrelin Acutely Increases Cardiac Perfusion Through a Nitric Oxide-Related Mechanism in Female Anesthetized Pigs Endocrinology. 2016 Jun;157(6):2403-15. doi: 10.1210/en.2015-1922. Epub 2016 Apr 21; Elena Grossini, Divya Garhwal, Sakthipriyan Venkatesan, Daniela Ferrante, Angelica Mele, Massimo Saraceno, Ada Scognamiglio, Jessica Mandrioli, Amedeo Amedei, Fa-bi-ola De Marchi, Letizia Mazzini. The Potential Role of Peripheral Oxidative Stress on the Neurovascular Unit in Amyotrophic Lateral Sclerosis Pathogenesis: A Preliminary Re-port from Human and In Vitro Evaluations Biomedicines. 2022 Mar 17;10(3):691. doi: 10.3390/biomedicines10030691).
In addition, it should be emphasized that in the procedure we conducted to carry out the NO dosage through this method, we executed deproteinization, as recommended (Andrea Brizzolari, Michele Dei Cas, Danilo Cialoni, Alessandro Marroni, Camillo Mo-rano, Michele Samaja, Rita Paroni, Federico Maria Rubino. High-Throughput Griess Assay of Nitrite and Nitrate in Plasma and Red Blood Cells for Human Physiology Studies under Extreme Conditions Molecules. 2021 Jul 28;26(15):4569. doi: 10.3390/molecules26154569). Moreover, the R values of the calibration standard curves, were settled on 0.99, which demonstrated the accuracy of our measurements.
We have also added some sentences about the above issues, as possible limitations (discussion, page 13): “Moreover, the fact that in this study we used the same methods for the assessment of plasma oxidative stress and NO previously adopted in the previous one, which was focused on the evaluation of the same markers in ALS patients at T0 (before ALCAR treatment), allowed us come to more meaningful conclusions about the effects of ALCAR treatment. Anyway, it could be useful to broaden that analysis of oxidative stress markers and NO by adding the quantification of other variables and using different analytical methods”.
Comment: Why do you use the Mitochondrial ROS Detection Assay Kit, which stains mitochondria-derived ROS, and not other methods that specifically measure cellular generation of H2O2 and O2.-?
Reply: As reported in the Introduction (page 2) “Many experimental findings showed that ALCAR could protect mitochondria against oxidative stress [18]. About this issue, ALCAR administration was able to induce mitochondrial biogenesis in hypoxic rats [19] and to increase mitochondrial mass after spinal cord injury [20]. In neurons and ALS animal models, ALCAR was found to exert protective effects through the modulation of mitochondrial function and neurotrophic activity [21]”.
In addition, in the Discussion (page 11), we say that “It is to note that changes in mitochondria morphology have been shown in neurons and glial cells from ALS patients and animal models [43,44]. Furthermore, it is to note that these alterations were also observed in both SOD1 and TDP43 ALS mice, indicating that they are common denominators of different genetic forms of ALS [45,46]. Those changes could lead to a cascade of events capable of altering mitochondrial respiration and ATP production, thus causing an increase in oxidative stress [47]”.
Since we have found increased oxidative stress in plasma of ALS patients, since mitochondria play a central role in the redox state balance and in ALS, as well, as above reported, in this study we wanted to analyze, in particular, the effects of plasma on mitochondrial ROS release, in HUVEC and astrocytes.
We have added a sentence about this issue, now (discussion, page 12): “About this issue, we used the Mitochondrial ROS Detection Assay since we were focused on the analysis of the mitochondrial function of HUVEC and astrocytes”.
In addition, as reported for the other assays, we have used this method in the previous paper about ALS (Elena Grossini, Divya Garhwal, Sakthipriyan Venkatesan, Daniela Ferrante, Angelica Mele, Massimo Saraceno, Ada Scognamiglio, Jessica Mandrioli, Amedeo Amedei, Fa-bi-ola De Marchi, Letizia Mazzini. The Potential Role of Peripheral Oxidative Stress on the Neurovascular Unit in Amyotrophic Lateral Sclerosis Pathogenesis: A Preliminary Re-port from Human and In Vitro Evaluations Biomedicines. 2022 Mar 17;10(3):691. doi: 10.3390/biomedicines10030691), which was focused on the evaluation of the effects of plasma of ALS patients at T0 (before ALCAR treatment), on HUVEC and astrocytes.

Reviewer 2 Report
The article entitled “Effects of Acetyl-L-carnitine on oxidative stress in Amyotrophic Lateral Sclerosis patients: evaluation on plasma markers and members of the neurovascular unit” is well written and offers a very interesting point of view about the possible pathogenesis of ALS, focusing on the role of oxidative stress and the potential role of the unit neurovascular. The methods employed to study the viability of endothelial vascular cells and astrocytes treated with plasma from ALS patients can be implemented as a very useful technique for biochemical diagnosis.
Minor changes would improve the article:
- Line 23 and 24: TBARS and GSH need to be defined.
- Line 56: remove a space before “can explain”.
- Line 63: this sentence “Anti-apoptotic effect of acetyl-l-carnitine and I-carnitine in primary cultured neurons” is not completed, and “L-carnitine” is repeated. Revise what you want to mean.
- Line 88: the “n” for the control group is 5, in opposite to the ALS group that is of 32 at T0 (another more for T1 and T2). Could it be possible to inform why this difference between ALS patients group and control group?, and if there is a guarantee of not being a possible sesgo at this disproportionate comparison? Another question is if you could inform about the inclusion criteria for recruited cases as control group, and the mean for age, and sex of this group.
- Line 96: Although T0 was defined in the abstract and in line 108, in this line 96 of the material and methods T0 should be established before.
- Line 103: the name of the genes “C9orf72, SOD1, TARDBP, and FUS” in cursive.
- Line 173: in figure 1, a legend or text to explain the “In vitro experimental protocol” might be of interest.
- Line 243: when you define the number of patients recruited for the study, it is no clear the total number, because in line 247, you specify the number in relation to the phase of the study, so consider if this initial numer should include the patients of all the phases (T0+T1+T2).
- Line 289: leave one space before “and nitric oxide”; and define in the legend “HUVEC”.
- Line 270, 304: Define in legend “ALCAR treatment”.
- Line 406: It would be very interesting that action mechanism of Carnitine be explained
- Line 409: in the sentence “all patients were taking not only ALCAR but also riluzole”, and joining it to the sentence in line 411-2: “it is not possible to hypothesize a control arm without riluzole”, one question can be made: it could be possible, however, a control with riluzole and without carnitine?, in order to establish a clear relationship of the additional effects of carnitine over riluzole in ALS patients.
- In all references, apport the DOI, and apply the indications by MDPI for the references.
Author Response
REVIEWER #2:
Comment:
- Line 23 and 24: TBARS and GSH need to be defined
- Line 56: remove a space before “can explain”
- Line 63: this sentence “Anti-apoptotic effect of acetyl-l-carnitine and I-carnitine in primary cultured neurons” is not completed, and “L-carnitine” is repeated. Revise what you want to mean
Reply: We thank the referee for the comment. We corrected as suggested.
Comment:
- Line 88: the “n” for the control group is 5, in opposite to the ALS group that is of 32 at T0 (another more for T1 and T2). Could it be possible to inform why this difference between ALS patients group and control group?, and if there is a guarantee of not being a possible sesgo at this disproportionate comparison? Another question is if you could inform about the inclusion criteria for recruited cases as control group, and the mean for age, and sex of this group.
Reply: We thank the referee for the possibility to better explain this point. In the first our manuscript in this regard we observed only minimal variability among controls, which justified us in limiting this cohort to 5. Furthermore, the objective of this study is to longitudinally evaluate the trend of plasma and cellular parameters within the same patient (comparing only baseline data with controls).
Also, as you suggested, we added the median age (65.5 (IQR: 54-71)) and sex (3M, 2F) in the manuscript.
Comment:
- Line 96: Although T0 was defined in the abstract and in line 108, in this line 96 of the material and methods T0 should be established before
- Line 103: the name of the genes “C9orf72, SOD1, TARDBP, and FUS” in cursive
Reply: We thank the referee for the comment. We corrected as suggested.
Comment:
- Line 173: in figure 1, a legend or text to explain the “In vitro experimental protocol” might be of interest.
Reply: hoping that can help in clarifying the methods, we have changed Figure 1, now.
Comment:
- Line 243: when you define the number of patients recruited for the study, it is no clear the total number, because in line 247, you specify the number in relation to the phase of the study, so consider if this initial numer should include the patients of all the phases (T0+T1+T2).
Reply: the total number of included patients is 32. The patients’ number is reduced over time due to patients’ drop-out (in a “normal” percentage considered the disease progression). So, from 32 patients at T0 we had 27 at T1 and 21 at T2.
Comment:
- Line 289: leave one space before “and nitric oxide”; and define in the legend “HUVEC”
- Line 270, 304: Define in legend “ALCAR treatment”
Reply: We thank the referee for the comment. We corrected as suggested.
Comment:
- Line 406: It would be very interesting that action mechanism of Carnitine be explained
Reply: We have added some sentences about this issue (discussion, page 13): “As regarding the effects on oxidative stress and mitochondria, it was shown that ALCAR can be metabolized in neuronal mitochondria to free carnitine and acetyl-CoA (De Marchi F, Venkatesan S, Saraceno M, Mazzini L, Grossini E.Acetyl-L-carnitine and Amyo-trophic Lateral Sclerosis: current evidence and potential use.CNS Neurol Disord Drug Targets. 2023 Mar 30. doi: 10.2174/1871527322666230330083757). The latter can be used as a substrate for lipid and neurotransmitters synthesis. Instead, free carnitine can be turned into products, such as carnitine derivatives of acyl-CoA conjugates, in the mitochondrial matrix, which could represent a valuable tool to reduce toxicity in oxidative stress conditions, though the prevention of accumulation of long chain fatty acids and long chain acyl-CoAs (Gustavo C Ferreira, Mary C McKenna. L-Carnitine and Ace-tyl-L-carnitine Roles and Neuroprotection in Developing Brain Neurochem Res. 2017 Jun;42(6):1661-1675. doi: 10.1007/s11064-017-2288-7. Epub 2017 May 16). Furthermore, free carnitine has been reported to play a key role for the mitochondrial fundtions, the fatty acid metabolism, and the production of ATP (Mohamed Ashraf Virmani, Maria Cirulli. The Role of l-Carnitine in Mitochondria, Prevention of Metabolic Inflexibility and Dis-ease Initiation. Int J Mol Sci. 2022 Feb 28;23(5):2717. doi: 10.3390/ijms23052717).
Moreover, many experimental findings demonstrated that ALCAR could protect mitochondria against oxidative stress. Also, ALCAR administration induced mitochondrial biogenesis in hypoxic rats and increased mitochondrial mass after spinal cord injury (De Marchi F, Venkatesan S, Saraceno M, Mazzini L, Grossini E.Acetyl-L-carnitine and Amyotrophic Lateral Sclerosis: current evidence and potential use.CNS Neurol Disord Drug Targets. 2023 Mar 30. doi: 10.2174/1871527322666230330083757). Considering what above reported, our data would, thus, corroborate the previous ones regarding the antioxidant effects of ALCAR, which would be related to the modulation of mitochondrial function”.
Comment:
- Line 409: in the sentence “all patients were taking not only ALCAR but also riluzole”, and joining it to the sentence in line 411-2: “it is not possible to hypothesize a control arm without riluzole”, one question can be made: it could be possible, however, a control with riluzole and without carnitine?, in order to establish a clear relationship of the additional effects of carnitine over riluzole in ALS patients.
Reply: Thanks for the possibility to better clarify this point. As a clinical practice, in our Centre, we start riluzole first and then ALCAR in the first month after diagnosis. Being a well-tolerated molecule, very few patients suspend it. Currently we have not performed analyzes on patients treated only with riluzole but it is certainly an excellent starting point for a future study. We better explained this point in the limitation paragraph.
Comment:
- In all references, apport the DOI, and apply the indications by MDPI for the references.
Reply: we adequated the reference style to the journal.

Reviewer 3 Report
ALS is a devastating disease with no effective treatments. As the authors indicate, although a portion of cases are genetic, the vast majority are sporadic and the precise causes are unknown. Thus, further information on disease markers in ALS patients as well as effects of possibly beneficial treatments are very much needed. This manuscript provides evidence both for increases in markers of redox imbalance in the plasma of ALS patients and the beneficial effects of acetyl carnitine (ALCAR) on this imbalance. In addition, the authors look at the impact of control and ALS plasma on astrocyte and endothelial cell survival, markers of mitochondrial function and NO release. The results are clearly presented and appropriately interpreted. However, it is critical to know whether any of these changes caused by ALCAR correlated with improvements in functional markers of ALS including ALSFRS score and FVC%. The Methods section states that clinical data was collected at each time point so the authors must have this information. This is particularly important because the effects of ALCAR on NO levels both in the plasma and the cell culture experiments appear either to be minimal (patient plasma) or negative (cell culture). If ALCAR is not very effective at treating the clinical symptoms of ALS despite partially restoring redox balance, then it would suggest that modulating NO in a cell type specific manner needs to be further evaluated in the context of ALS. In addition, the authors should provide information on whether the patients had a genetic or sporadic form of ALS and whether there were differences in the oxidative stress markers at baseline between the two groups as well as in the response to ALCAR treatment. This information would be useful in helping to understand differences in the two forms of the disease.
The English could be improved as the phrasing and word use is problematic throughout the manuscript.
Author Response
REVIEWER #3:
Comment: it is critical to know whether any of these changes caused by ALCAR correlated with improvements in functional markers of ALS including ALSFRS score and FVC%. The Methods section states that clinical data was collected at each time point so the authors must have this information.
Reply: thanks for the positive opinion about our paper and the possibility to improved it. As suggested, we added in the Results section the ALFRS-R, FVC% and BMI values: “At T1 the median ALSFRS-R was 37 (IQR: 30.50-41.00), with an FVC% of 72 (IQR: 55-95) and BMI of 22.25 (IQR: 19.98-25.88). At T2 the median ALSFRS-R was 33 (IQR: 31.00-38.00), with an FVC% of 69 (IQR: 51-90) and BMI of 23.45 (IQR: 19.93-26.88)”. We explain the relatively stabilization of FVC% and BMI progression between T1 and T2 related to the drop-out of patients with more severe phenotype.
Comment: This is particularly important because the effects of ALCAR on NO levels both in the plasma and the cell culture experiments appear either to be minimal (patient plasma) or negative (cell culture). If ALCAR is not very effective at treating the clinical symptoms of ALS despite partially restoring redox balance, then it would suggest that modulating NO in a cell type specific manner needs to be further evaluated in the context of ALS.
Reply: we agree with the Reviewer and have added a sentence as limitation about this issue (discussion, page 13): “In addition, it could be useful to investigate the intracellular pathways implicated in plasma effects in HUVEC and astrocytes, particularly in relation to NO release”.
Comment: In addition, the authors should provide information on whether the patients had a genetic or sporadic form of ALS and whether there were differences in the oxidative stress markers at baseline between the two groups as well as in the response to ALCAR treatment. This information would be useful in helping to understand differences in the two forms of the disease.
Reply: in our cohort we had 6 patients with the C9Orf72 mutation (18.75%, in line with the literature prevalence of this mutation). No difference was observed at baseline and over disease course in regard of the oxidative stress markers. We added a sentence on Results paragraph about this point.

Round 2
Reviewer 1 Report
None of the issues raised in my previous reviewing have been solved properly.
My detail comments were in my first reviewing. The problem with this ms is that the authors decided not to do any further experimental work. They simply tried to defend their position by stating that others followed the same methodology. I was clear explaining that the methodology they used had mistakes and, consequently, the conclusions were not supported based on the results obtained.Minor editing issues.
Author Response
The randomized double-blind placebo-controlled trial of acetyl-L-carnitine for ALS published on
2013 showed that this molecule had options in ALS therapy (Amyotroph Lateral Scler
Frontotemporal Degener. 2013 Sep;14(5-6):397-405). In fact, following these positive findings, the
authors concluded that a phase III trial was needed. Based on the grim prognosis of ALS and the
lack of effective treatments (still an unmet need), it is quite surprising that no clinical trial on ALCAR
and ALS has been set up since then (I could not find any at www.clinicaltrial.gov). This referee
would like to know why.
Regarding the methodology:
- As clearly noted by Wade and van Rij (Clin Chem. 1989;35:336) most of problems associated
with TBA assay were ignored by many researchers. Despite of these points, MDA is still used as an
oxidative stress biomarker. It is now well accepted by any expert that this assay is not reliable as a
marker of oxidative stress.
- Plasma GSH levels were measured on plasma samples through the Glutathione Assay Kit 136
(Cayman Chemical). This is another methodological mistake since this assay does not accurately
prevent GSH oxidation. See e.g. Anal Biochem. 1994 Mar;217(2):323-8.
- Plasma NO were quantified in plasma samples by the Griess system and expressed as nitrite
production. However, particularly in plasma samples, this methodology is not accurate. The
sensitivity of the flow injection analysis, compared to the traditional Griess method, is higher by a
factor of 500. It has high sensitivity, high reproducibility, and low susceptibility to interference
(Nitric Oxide. 1999 Jun;3(3):225-34).
- Why do you use the Mitochondrial ROS Detection Assay Kit, which stains mitochondria-
derived ROS, and not other methods that specifically measure cellular generation of H2O2 and O2.-
?
We performed new analyses in order to address the points raised by the Reviewer.
In particular, we have measured the plasma 4-HNE levels (through an ELISA assay), the plasma
GPX activity (by means of the Cayman assay which measures the activity of GPx through a couple
reaction with the GSH reductase), and the plasma NO, as both nitrites and nitrites/nitrates (NOx),
through a fluorimetric assay.
New paragraphs have been added in Abstract (page 1, lines 25-27), Materials (page 4, lines 139-151,
172-177; page 5, lines 178-197), Results (page 8, lines 292-302; page 14, lines 373, 378), Discussion
(page 14, lines 385, 387, 388-390, 398-400; page 15, lines 417-42, 443, 444; page 16, lines 465, 467-477;
page 17, lines 509-512, 534-537).
Also, we have added new graphs in Figure 2 and new references.
As specified in the revised manuscript, the new measurements confirmed previous data about the
lipid peroxidation and the antioxidants in the plasma of ALS patients. The 4-HNE levels were higher
in the plasma of ALS patients at almost all time points, although with a reducing trend. It is to note
that, as observed for the TBARS, also for the 4-HNE, differences were found between fast progressor
patients and slow progressors, that could indicate a stronger dysfunction in the oxidative stress in
patients with a more severe disease and, likely, a minor beneficial effect of the available treatment.
Our data on 4 -HNE are, therefore, in agreement with the previous ones, which had shown a
correlation between the aforementioned marker and a greater disease severity (Devos D, Moreau C,
Kyheng M, Garçon G, Rolland AS, Blasco H, Gelé P, Timothée Lenglet T, Veyrat-Durebex C, Corcia
P, Dutheil M, Bede P, Jeromin A, Oeckl P, Otto M, Meininger V, Danel-Brunaud V, Devedjian JC,
Duce JA, Pradat PF. A ferroptosis-based panel of prognostic biomarkers for Amyo-trophic Lateral
Sclerosis. Sci Rep. 2019 Feb 27;9(1):2918. doi: 10.1038/s41598-019-39739-5).
Also as regarding the GPx activity, the results obtained confirm those relating to GSH, since it was
much lower than that found in the plasma of the healthy controls at all time-points. Our data are in
agreement with previous observations found in the plasma of ALS patients and in postmortem brain
homogenates (Katarzyna Patrycja Dzik, Damian Józef Flis, Zofia Kinga Bytowska, Mateusz Jakub
Karnia, Wieslaw Ziolkowski, Jan Jacek Kaczor Swim Training Ameliorates Hy-perlocomotion of
ALS Mice and Increases Glutathione Peroxidase Activity in the Spinal Cord Int J Mol Sci. 2021 Oct
27;22(21):11614. doi: 10.3390/ijms222111614; Cova, E.; Bongioanni, P.; Cereda, C.; Metelli, M.R.;
Salvaneschi, L.; Bernuzzi, S.; Guareschi, S.; Rossi, B.; Ceroni, M. Time course of oxidant markers and
antioxidant defenses in subgroups of amyotrophic lateral sclerosis patients. Neurochem. Int. 2010,
56, 687–693;Przedborski, S.; Donaldson, D.; Jakowec, M.; Kish, S.J.; Guttman, M.; Rosoklija, G.; Hays,
A.P. Brain superoxide dismutase, catalase, and glutathione peroxidase activities in amyotrophic
lateral sclerosis. Ann. Neurol. 1996, 39, 158–165).
Finally, we analyzed the plasma NO, as both nitrites and total nitrites and nitrates, now. In order to
perform an accurate measurement, the plasma samples were ultrafiltered through the Amicon®
Ultra filter (30kDa MWCO Merck KGaA, Darmstadt, Germany), in order to remove proteins, as
suggested. The data we obtained were compared with those of standard curves for both nitrites and
NOx.
As reported in the text, the plasma nitrites and NOx levels were much lower than those found in the
healthy controls, from T0 to T2. Our data, therefore, highlighted the presence of an endothelial
dysfunction in ALS patients, which was not affected by the treatment with ALCAR. Those data were
confirmed by the in vitro experiments in HUVEC.
We hope to have addressed the Reviewer's requests, now.
We have also included in the limits of the study the possibility of expanding the quantifications of
oxidants by adding hydrogen peroxide and superoxide and the measurement of NO through flow
injection analysis.